# Plasma Glycosaminoglycans in Children with Juvenile Idiopathic Arthritis Being Treated with Etanercept as Potential Biomarkers of Joint Dysfunction

**DOI:** 10.3390/biomedicines10081845

**Published:** 2022-07-31

**Authors:** Magdalena Wojdas, Klaudia Dąbkowska, Kornelia Kuźnik-Trocha, Grzegorz Wisowski, Iwona Lachór-Motyka, Katarzyna Komosińska-Vassev, Krystyna Olczyk, Katarzyna Winsz-Szczotka

**Affiliations:** 1Department of Clinical Chemistry and Laboratory Diagnostics, Faculty of Pharmaceutical Sciences in Sosnowiec, Medical University of Silesia, ul. Jedności 8, 41-200 Sosnowiec, Poland; klaudia_092@vp.pl (K.D.); kkuznik@sum.edu.pl (K.K.-T.); vis@sum.edu.pl (G.W.); kvassev@sum.edu.pl (K.K.-V.); olczyk@sum.edu.pl (K.O.); winsz@sum.edu.pl (K.W.-S.); 2Department of Rheumatology, The John Paul II Pediatric Center in Sosnowiec, ul. G. Zapolskiej 3, 41-218 Sosnowiec, Poland; iwlamo@tlen.pl

**Keywords:** juvenile idiopathic arthritis, extracellular matrix, etanercept, cartilage turnover biomarkers, matrix metalloproteinases

## Abstract

We assessed the effect of two-year etanercept (ETA) therapy on the metabolism of the cartilage extracellular matrix (ECM) in patients with juvenile idiopathic arthritis (JIA). Methods: We performed a quantitative evaluation of glycosaminoglycans (GAGs) (performed by the multistage extraction and purification method) in blood obtained from patients before and during 24 months of ETA treatment, as potential biomarker of joint dysfunction and indicators of biological effectiveness of therapy. Since the metabolism of GAGs is related to the activity of proteolytic enzymes and prooxidant–antioxidant factors, we decided to evaluate the relationship between GAGs and the levels of metalloproteinases (MMP), i.e., MMP-1 and MMP-3 (using immunoenzymatic methods), as well as the total antioxidative status (TAS) (using the colorimetric method) in blood of the JIA patients. Results: When compared to the controls, GAGs and TAS concentrations were significantly lower in patients with an aggressive course of JIA qualified for ETA treatment. MMP-1 and MMP-3 levels were significantly higher versus control values. An anti-cytokine therapy leading to clinical improvement does not lead to the normalization of any of the assessed parameters. GAGs concentration is significantly related to MMP-1, MMP-3, TAS, TOS, and CRP levels. Conclusion: The results of the present study indicate the necessity of constant monitoring of the dynamics of destructive processes of articular cartilage in children with JIA. We suggest that GAGs may be a useful biomarker to assess the clinical status of the extracellular matrix of joints.

## 1. Introduction

Juvenile idiopathic arthritis (JIA) is the most common form of inflammatory systemic connective tissue disease in children, with a multifactorial pathogenesis and varied course. It is assumed that the disorder is more frequently observed in genetically predisposed children, exposed to adverse environmental factors [1,2]. These factors, as a result of disturbed immunological mechanisms, lead to hyperactivation of proinflammatory cytokines. Tumor necrosis factor alpha (TNF-α), next to interleukin (IL) 1 or IL-6, is one such cytokine. Prolonged and uncontrolled activation of the mentioned proinflammatory mediator, by the activation of other cytokines and chemokines expression, promotion of angiogenesis or suppression of regulatory T cells, leads to chronic inflammation, which induces changes in the structure and function of the joints [3,4]. Persistent inflammation of the synovial membrane of a joint with subsequent formation of invasive synovial tissue develops at different rates and may lead to destruction of cartilage and subchondral bone tissue and periarticular tissues, resulting in disability of the JIA patient [5]. The progressive wear of joint structures is associated with disturbances in the metabolism of the extracellular matrix (ECM) components. The ECM is a multicomponent, dynamic-network structure with gel-like properties, especially in the cartilage, with a composition and arrangement that determine its mechanical functions and immunological characteristics. Among ECM compounds, proteoglycans (PGs), co-formed by glycosaminoglycan (GAGs) chains, play a special role in maintaining the mechanical–immunological properties of cartilage [6,7]. Under physiological conditions, PGs co-create the dynamic structure of cartilage ECM, which undergoes continuous remodeling, and its disorders, reflected by GAGs concentration in blood, which may be a result of excessive activity of matrix metalloproteinases (MMPs). MMPs are involved in the alteration of ECM macromolecules including PGs. Degradation of ECM components, besides ECM remodeling, eliminates biophysical and structural barriers that constitute an effective mechanism regulating various cell behaviors, from proliferation, adhesion, and migration to their differentiation and death, which occurs both in normal and pathological conditions [8,9]. Increased activity of MMPs has been found in biological fluids from JIA patients treated with methotrexate [10]. However, little is known about their association with the metabolism of PGs in children with JIA undergoing anti-TNF therapy. We cannot exclude the possibility that prooxidant–antioxidant imbalance is another pathogenetic factor leading to impaired PGs/GAGs metabolism in children with JIA. It is known that reactive oxygen species (ROS) are involved in the extracellular degradation of ECM components. On the other hand, children with JIA are known to suffer from oxidative stress manifested by increased free radical activity and impaired antioxidant systems [11,12]. Apart from excessive proteolysis, dysfunction of joint components may be favored by impaired processes of their biosynthesis, which is regulated by transforming growth factor β1 (TGF-β1) [13].

In view of the above as well as the clinical consequences of incorrectly or late-diagnosed and -treated JIA, contributing to the permanent pathological changes in patients’ locomotor system, the aim of the paper is to assess the dynamics of changes in GAGs concentrations, as potential biomarkers of joint dysfunction and indicators of biological effectiveness of therapy, in the blood of patients with JIA, both before and during 24-month treatment with etanercept (ETA), contributing to clinical improvement in patients. Since the metabolism of GAGs is related to the activity of proteolytic enzymes and prooxidant–antioxidant factors, we decided to evaluate the relationship between circulating glycans and the levels of MMP-1 and MMP-3, i.e., metalloproteinases involved in the depolymerization of PGs core proteins, as well as total oxidative potential (TOS, total oxidative status) and total antioxidant capacity (TAS, total antioxidative status) in blood of the patients. Since the concentration of GAGs in blood is related to the amount of their tissue synthesis, we also decided to evaluate their relationship with TGF-β1. TOS and TGF- β1 results have been published in a previous report [14].

## 2. Materials and Methods

### 2.1. Patients and Samples

The biological material for the research, i.e., blood samples, was obtained from 38 Polish children of both sexes in the age range of 4–11 years. Children were diagnosed with JIA according to the criteria of the International League of Associations for Rheumatology, based on disease duration and clinical symptoms—including joint pain and swelling, joint mobility limitations, and growth abnormalities [15]. Patients qualified for the research were treated with ETA in the Department of Rheumatology at John Paul II Pediatric Center in Sosnowiec. An exclusion list was used to qualify subjects for the study. Moreover, accuracy of the diagnosis was confirmed by laboratory tests. Among the diagnostic tests performed on JIA patients were indicators of the inflammatory response, i.e., C-reactive protein (CRP, immunonephelometric assay) and erythrocyte sedimentation rate (ESR, Westergren method), rheumatoid factor (latex enhanced immunoturbidimetric test), and the titer of antinuclear antibodies (indirect immunofluorescence assay).

Disease activity was assessed in all patients according to Juvenile Arthritis Disease Activity Score-27 (JADAS-27). The JADAS-27 (range 0–57) was calculated by summing the scores of criteria. The latter include physician’s global assessment of disease activity (PGA) on a 10 cm visual analogue scale (VAS); parent/patient global assessment of well-being on a 10 cm VAS; active arthritis, defined as joint swelling or limitation of movement accompanied by pain and tenderness, assessed in 27 joints; and ESR. The patients in whom Methotrexate (MTX, at ≤15 mg, calculated per square meter of body surface area), Sulfasalazine (SSA, at 30 mg calculated per kilogram of body weight), and Encorton (EC, at a maximum dose of 1 mg, calculated per kilogram of body weight) therapy failed to improve clinical status or was not tolerated were qualified for the study. It was the basis for the implementation of ETA therapy in these patients. In Poland, biological drugs, including ETA, for rheumatic diseases are available within the so-called Therapeutic Programs (all of the JIA patients participated in Therapeutic Program employing TNF blockers, i.e., B.33). Patients were classified for biological treatment with ETA in case they suffered from the following types of JIA: polyarticular JIA or oligoarticular JIA (extended and persistent), with poor prognosis factors. The polyarticular form of JIA is identified in patients with at least 5 swollen joints and at least 3 joints with limited mobility, high ESR or CRP values, and a physician’s assessment of disease activity for at least 4 points on a 10-point scale. The characteristic features of the oligoarticular form of JIA are the presence of at least 2 joints swollen or with limited mobility and a physician’s assessment of disease activity of at least 5 on a 10-point scale, with accompanying pain and/or tenderness. Patients with other than polyarticular and oligoarticular forms of JIA, with a history of injury or surgery to the musculoskeletal system within the last 3 years, with autoimmune and metabolic diseases, including diabetes, cancer, kidney disease, liver disease, and chronic infections, were not included in the research.

ETA was administered by subcutaneous injection. ETA was used twice a week at intervals of 3–4 days at a dose of 0.4 m/kg body weight (up to a maximum dose of 25 mg) or 0.8 mg/kg body weight (up to a maximum dose of 50 mg) once a week. In all patients, ETA was used together with the MTX, SSA, and EC. After 3 months of effective therapy, both SSA and EC were withdrawn.

In all JIA patients, the assessment of circulating biomarkers of ECM alterations in the blood was performed both before the initiation of biological therapy (T0), and in the same patients at the following intervals, i.e., after the third (T3), the sixth (T6), the twelfth (T12), the eighteenth (T18), and the twenty-fourth (T24) month of ETA treatment, i.e., after clinical improvement. ACR criteria were used to assess clinical improvement [16].

As control, blood samples obtained from 30 healthy children in comparable age range to the group of JIA patients were used. Children in the control group, included in our study, did not suffer from any diseases that required hospitalization and did not undergo surgical procedures during the previous year. Moreover, they were not treated pharmacologically just before the studies, and their laboratory test results showed no deviations from the suggested reference values. The clinical data of healthy individuals and JIA patients enrolled in our study are shown in Table 1.

Venous blood samples were collected between 7.00 and 9.00 a.m. after overnight fasting. Citrate-treated tubes were used to extraction and determination of plasma GAGs and heparin-treated tubes for measuring plasma MMP-1, MMP-3, and TAS levels. The plasma obtained both from JIA patients and control group were divided into portions and stored at −80 °C until the initiation of the study.

All subjects gave their informed consent for inclusion before they participated in the study. The study was conducted in accordance with the Declaration of Helsinki, and the protocol was approved by the Local Bioethics Committee of the Medical University of Silesia in Katowice (KNW/0022/KB/168/18).

### 2.2. Extraction and Determination of Plasma GAGs

The determinations of plasma GAGs were performed by the multistage extraction and purification using papaine hydrolysis and alkali elimination, according to the Volpi et al. [17] and Olczyk et al. [18] method previously described [19].

The total amount of GAGs was quantified as a hexuronic acid by the carbazole methods of Volpi et al. [20] and Filisetti-Cozzi and Carpita [21] as well as van den Hoogen et al. [22]. In the first stage of hexuronic acid concentration determination, ammonium amidosulfonate solution was added to aqueous solutions of GAGs. Then, hydrolysis of GAGs to their monosaccharide components with simultaneous conversion of glucuronic and/or iduronic acid residues to the corresponding furan derivative was carried out using concentrated sulfuric acid containing sodium tetraborate solution. The previously mentioned reaction was carried out at 100 °C for 10 min. Then, after cooling the samples, their absorbance was first measured at 525 nm. In the next step, furan derivatives in the analyzed samples were conjugated with carbazole dissolved in 99% ethanol. The above reaction was carried out by heating samples at 100 °C for 15 min. After cooling, the absorbance of the test samples was measured again at 525 nm. The difference in absorbance values obtained during the measurements was used for calculations, and the concentration of hexuronic acids was interpolated from the corresponding calibration curve, plotted for standard solutions of D(+)glucuronolactone, with concentrations ranging from 0 to 70 µg/mL. The total amount of glucuronic acid was determined with analytical sensitivity of 0.5 mg/L. The intra-assay variability was less than 10%.

### 2.3. Assay of the Concentration of MMP-1 and MMP-3

MMP-1 and MMP-3 levels were measured in duplicate, using blindly tested coded plasma samples. Determination of concentrations of both MMPs was completed within a day. Thus, the inter-assay variation was insignificant. Enzyme-linked immunosorbent assays (ELISA) were used, in accordance with the protocol of the manufacturer. We used ELISA kits dedicated to scientific purposes. Plasma concentrations of MMP-1 and MMP-3 were determined with ELISA Kits by Cloud-Clone Corp. (Houston, TX, USA), with a minimum detection of 0.061 ng/mL (MMP-1) and 1.8 ng/mL (MMP-3), respectively. The mentioned assays have high sensitivity and excellent specificity for detection of MMP-1 and MMP-3, respectively. No significant cross-reactivity or interference between MMP1 or MMP-3 and analogues was observed. For all parameters tested, the intra-assay variability was less than 10%.

### 2.4. Assay of the Concentration of TAS

Total antioxidative status levels were measured using a photometric test system ImAnOx kit, supplied by Immundiagnostik AG (Bensheim, Germany), according to the method previously described [10]. The minimal detectable concentration was 130 μmol/L. The intra-assay CV was lower than 8%.

### 2.5. Statistical Analysis

Statistical analysis was performed using Statistica 13.3 package (StatSoft, Krakow, Poland). Normality of distribution was checked by Shapiro–Wilk test. Homogeneity of variance was assessed by Levene’s test. The data obtained were expressed as mean values and standard deviation for the variables normally distributed and as median and upper and lower quartile for variables with skewed distribution. Since the variables were normally distributed, the parametric Student’s t-test was used to evaluate the differences between untied variables. For not normally distributed variables, Mann–Whitney U test was used. We used one-way analysis of variance ANOVA with repeated measures and Tukey’s post hoc test to compare the same parameters in each patient before and during ETA therapy. We employed Pearson’s correlation coefficient, modified by Bonferroni’s multivariate correction, for the statistical analysis of correlations between two variables. For each statistical analysis, a significance level of *p* < 0.05 was considered.

## 3. Results

The results regarding the evaluation of GAGs, MMP-1, MMP-3, and TAS were analyzed only in JIA patients who completed the whole 24-month TNF-α therapy. The results are presented in Table 2. The results regarding the evaluation of TOS, TGF-β1, and CRP, used for correlation analysis, are included in Table 1.

### 3.1. Plasma Levels of GAGs in Healthy Children and JIA Patients

The results of the study revealed a statistically significant (*p* = 0.000031) decrease in plasma GAGs concentration in patients qualified for ETA treatment (T0), as compared to the concentration of the molecules in plasma of healthy children. It was demonstrated that a two-year treatment with ETA, which contributes to clinical improvement in patients, at the same time does not lead to normalization of the glycans metabolism. GAGs concentration in patients’ blood after 24 months of ETA therapy was still significantly (*p* = 0.000000) lower in comparison to the plasma glycans concentration characterizing healthy children.

### 3.2. Plasma Levels of MMP-1, MMP-3, and TAS in Healthy Children and JIA Patients

The results of the study revealed a statistically significant increase in plasma MMP-1 (*p* = 0.002669) and MMP-3 (*p* = 0.000004) concentration in patients qualified for ETA treatment (T0), as compared to the concentration of the molecules in plasma of healthy children. It was also reported that biological treatment did not contribute to the normalization of blood concentrations of the enzymes. Plasma concentrations of MMP-1 (*p* = 0.000298) and MMP-3 (*p* = 0.004699), characterizing patients after two years of ETA therapy, were still significantly higher in comparison to concentrations of the evaluated parameters in the blood of healthy children. The quantitative analysis of TAS levels in the blood of untreated JIA patients showed a significant decrease (*p* = 0.000000) in its concentration versus control values. The treatment promoted a significant increase in TAS concentration. Thus, the average TAS concentration in treated JIA patients was significantly higher (*p* = 0.005012) than the value reported in the blood of healthy children.

### 3.3. Changes in Plasma Levels of GAGs, MMP-1, MMP-3, and TAS in Patients with JIA during ETA Treatment

The levels of the evaluated parameters in the patients’ blood samples collected before starting ETA therapy (T0) and after 3 (T3), 6 (T6), 12 (T12), 18 (T18), and 24 (T24) months of ETA application are presented in Figure 1.

As Table 2 indicates, two trends of changes in plasma concentrations of the GAGs were demonstrated during biological treatment of patients with JIA. The first, gradual upward trend was observed up to the 6th month of treatment (T6), while continued use of ETA in patients led to a gradual decrease in circulating GAGs concentrations. The highest concentration of GAGs, reported at the sixth month of therapy (T6), is statistically different from the concentration of GAGs in the group of patients before ETA treatment (T0) (*p* = 0.001110). The lowest concentration of GAGs was observed after 24 months of therapy. Statistical analysis of the obtained results revealed significant differences in GAGs concentration between patients from the T24 group and, respectively, the T3 (*p* = 0.000023), T6 (*p* = 0.000020), and T12 (*p* = 0.003953) groups and between patients from the T18 group and the T3 (*p* = 0.000110), T6 (*p* = 0.000020), and T12 (*p* = 0.035615) groups.

In addition, similar trends of changes in MMP-1 and MMP-3 blood concentrations were demonstrated in patients with JIA during ETA therapy. Similarly to GAGs, i.e., the first, upward trend was observed until the 6th month of treatment (T6), whereas the continuation of ETA therapy in patients with JIA led to a gradual decrease in plasma concentrations of circulating MMP-1 and MMP-3. Statistical analysis of the results revealed significant differences in MMP-1 levels between the group of patients before ETA treatment (T0) and patients from the following groups: T3 (*p* = 0.001959), T6 (*p* = 0.000021), and T12 (*p* = 0.004166). Moreover, significant differences in MMP-1 levels were also reported between T18 and T6 (*p* = 0.021970), T24 and T3 (*p* = 0.008983), T24 and T6 (*p* = 0.000032), and T24 and T12 (*p* = 0.017385). The highest MMP-3 levels obtained at 6 months of treatment (T6) were significantly different from the MMP-3 levels in the group of patients before therapy (T0) (*p* = 0.003547) and the T12 (*p* = 0.000023), T18 (*p* = 0.000021), and T24 (*p* = 0.000020) groups. Moreover, MMP-3 concentration in the group of children in the third month of ETA therapy (T3) was statistically significantly higher than the MMP-3 concentration in T12 (*p* = 0.002858), T18 (*p* = 0.000541), and T24 (*p* = 0.000190).

Changes in blood TAS levels in patients are characterized by an upward trend during ETA treatment. The lowest TAS concentration, reported in the T0 group, was significantly different from the TAS concentrations in the following patient groups: T6 (*p* = 0.000050), T12 (*p* = 0.000052), T18 (*p* = 0.000168), and T24 (*p* = 0.000020). The highest TAS concentration, reported in the T24 group, was significantly different from the TAS concentration obtained in the following groups: T3 (*p* = 0.000020), T6 (*p* = 0.000021), T12 (*p* = 0.000021), and T18 (*p* = 0.000020). Moreover, statistical analysis of results revealed significant differences in the TAS levels between T3 and T6 (*p* = 0.040508) and T3 and T12 (*p* = 0.041981).

### 3.4. Correlation Analysis between Plasma GAGs and MMP-1, MMP-3, TOS, TAS, TGF-β1, and CRP Levels in JIA Patients

As shown in Table 3, statistical analysis revealed a correlation between GAGs and MMP-1 levels in the blood of patients with JIA before ETA therapy (r = 0.768, *p* = 0.000), at the 3rd (r = 0.678, *p* = 0.000), 6th (r = 0.738, *p* = 0.000), 12th (r = 0.779, *p* = 0.000), 18th (r = 0.692, *p* = 0.000), and 24th (r = 0.693, *p* = 0.000) months of therapy. Furthermore, there was a significant correlation between plasma GAGs levels and MMP-3 levels in the untreated patients before ETA therapy (r = 0.386, *p* = 0.017) and at the 3rd (r = 0.614, *p* = 0.000), 6th (r = 0.664, *p* = 0.000), 12th (r = 0.701, *p* = 0.000), 18th (r = 0.720, *p* = 0.000), and 24th (r = 0.811, *p* = 0.000) months of therapy. The evaluation of the relationship between plasma GAGs concentrations and TAS levels observed in patients with JIA revealed the presence of significant correlations between the mentioned variables before the ETA treatment (r = −0.865, *p* = 0.000), as well as in the 3rd (r = 0.718, *p* = 0.000) and 6th (r = −0.917, *p* = 0.000) months of its duration. The analysis of the correlation between plasma GAGs concentration and TOS levels revealed the presence of significant correlations between the concentrations of the above-mentioned parameters in the 3rd (r = 0.753, *p* = 0.000), 12th (r = −0.791, *p* = 0.000), and 24th (r = 0.541, *p* = 0.000) months of ETA treatment. As a result of statistical analysis, no significant correlation was found between plasma GAGs levels and TGF-β1 concentration characteristic for patients before and during ETA therapy. The analysis of the correlation between plasma GAGs concentration and inflammatory process activity, expressed by serum CRP concentration, allowed for the observation of a significant relationship between the concentrations of the above-mentioned parameters in patients before the therapy (r = −0.769) as well as in the 12th (r = 0.439) month of its duration.

## 4. Discussion

GAGs form a dynamic structure of ECM of cartilage, which undergoes continuous remodeling. Disturbances in the synthesis, modification, or degradation of GAGs, which are one of the links in the pathogenetic chain of changes leading to the development of JIA, are reflected in changes in the concentration of GAGs in the blood of patients [23]. It was shown that in children with aggressive and refractory responses to standard therapy with disease-modifying drugs or steroids, who qualified for biological treatment, blood GAGs concentrations were significantly lower compared to the concentrations of these compounds characterizing healthy children. The tendency of changes in plasma GAGs concentration found in the present study cannot be verified by comparing our results with the results of other authors, because so far quantitative assessment of plasma GAGs concentration has not been reported in children with JIA qualified for biological treatment. Similarly to our research, decreased GAGs concentrations were found in blood, urine, and joint fluid of children with “newly” diagnosed, previously untreated JIA, and qualified for methotrexate therapy [24,25]. The authors indicated a decrease in blood and urine glycan concentrations due to low concentrations of sulfated GAGs, particularly chondroitin sulfates (CS). It has been shown that the concentrations of dermatan sulfates (DS), keratan sulfates (KS), and hyaluronic acid (HA) in the biological fluids of patients with arthropathy increase, while those of heparan sulfates and heparins do not change [24,26]. The profile of GAGs in body fluids observed in patients, according to the authors, reflects the different sensitivity of individual glycan fractions to the influence of factors leading to the development of JIA. The latter factors include proinflammatory cytokines and growth factors, including TNF-α, IL-1, IL-6, or TGF-β, which, being at the same time compounds stimulating GAGs biosynthesis, seem to regulate in particular the metabolism of DS, KS, and HA [4,27,28]. Whereas, CS transformations seem to be more related to the processes of tissue degradation of their proteoglycan linkages, occurring under the influence of specific proteases as well as ROS [29,30].

Similar observations concerning the decrease in glycan concentration in the synovial fluid of children with JIA, in relation to adult patients with osteoarthritis and healthy children, were demonstrated by Struglics et al. [31]. Despite the many clinical similarities between rheumatoid arthritis and JIA, it seems that the observed different trends in changes of circulating GAGs in patients are due to disturbances in the skeletal-metabolism characteristic of young patients [32]. It is known that during the growth phase, cartilage tissue undergoes particularly intensive processes of structural modeling, with formation processes predominating [33]. As a result of the dysfunction of the systems modulating cartilage metabolism in patients with arthropathies, anabolic metabolism is impaired relative to the extent of catabolic processes of joint structures [34,35]. Thus, the lower, relative to healthy subjects, plasma concentrations of GAGs characterizing patients with JIA reflect the abnormal tissue metabolism of the proteoglycan components of the ECM, likely occurring already at the early, preclinical stages of the disease. Although the early stages of ECM-component degradation processes should be reflected by an increase in blood GAGs concentrations, it appears that increased depolymerization, occurring over a prolonged period of time in children with JIA, especially in those with an aggressive form of arthropathy, may lead, through substrate depletion, to a gradual decrease in the plasma pool of these compounds. It has been suggested that the degradation processes of matrix glycoproteins occurring in the early stages of the disease are reversible and precede the degradation of the collagen network, whose disruption—in adult individuals with rheumatic diseases—is irreversible and leads to permanent impairment of joint function [36,37]. The reduction in circulating glycans concentrations observed in this study seems to confirm the above thesis and indicates that at the onset of clinical symptoms of the disease, the pool of tissue GAGs is significantly reduced, and the processes of synthesis of ECM components do not compensate the degradation of these compounds. Moreover, the “strength” of the degradation is not inhibited by the drugs previously used in the treatment of sick children, including SSA, EC, and MTX, which also did not contribute to the suppression of the inflammatory process in these patients.

The compounds directly involved in the cycle of proinflammatory incidents resulting in progressive degradation of joint structures in patients with JIA include specific proteolytic enzymes as well as non-specific ROS [11,29,38,39]. The main enzymes involved in the degradation of the protein core of PGs are MMPs [9,38]. The thesis of increased—stimulated by MMPs—degradation of PGs/GAGs in the course of JIA is confirmed by significantly higher concentrations of MMP-1 and MMP-3 in the blood of patients with active arthropathy, who qualified for ETA therapy. Moreover, the concentration of MMPs was significantly related to the plasma concentration of GAGs in these patients. The trend of changes in the blood concentrations of the MMPs in the patients with JIA observed in the present study is consistent with the observations of other researchers [10,40,41].

In addition to the excessive proteolysis stimulated by MMPs, the decreased concentration of GAGs in the blood of children with JIA seems to be a result of the systemic prooxidant–antioxidant imbalance, defined as oxidative stress [11,42]. As evidence of excessive ROS activity, we demonstrated in a previous report a significant increase in TOS in the blood of children with biologically untreated arthropathy [14]. The existence of a close relationship between the concentration of TOS and the total concentration of GAGs in the plasma of patients indicates the significant involvement of this free radical form of post-synthetic modification of molecules in the mechanism of ECM-structure change in the course of JIA. As shown in the present study, patients with aggressive forms of JIA, who qualified for biological therapy, are characterized by the depletion of the antioxidant reserves of the system. A reduction in total antioxidant potential observed in the blood of patients with untreated arthropathy indicates that the mechanisms of the individual antioxidant response are different. The results of the study have indicated that the low serum concentration of TAS was significantly related to the concentration of GAGs in patients’ blood. The demonstrated negative correlation between the variables seems to indicate that in the course of JIA, glycosaminoglycans, besides their structural and regulatory functions in relation to cellular interactions, play a significant antioxidant role. Thus, the low concentration of glycans in the blood of patients probably reflects the “depletion” of the tissue reservoir of these GAGs, resulting from the “receiving” of free radical attacks by these compounds. It has been proven that in some tissues, GAGs are the most effective antioxidants, and modifications in the composition of the ECM can increase susceptibility to oxidative stress. Similar mechanisms probably characterize the glycosaminoglycan content of articular cartilage. Indeed, GAGs, regardless of their tissue localization, are characterized by their ability to directly “scavenge” reactive molecules. Moreover, GAGs exhibit antioxidant capacity by chelating transition metals such as copper and iron, which, in turn, are responsible for initiating the Fenton reaction, which generates reactive hydroxyl radicals with high oxidative potential, especially toward lipids [43,44,45]. Lipid peroxidation intermediates can induce activation of both nuclear transcription factor kappa B (NF-κB), which plays a significant role in immune and inflammatory processes and caspases, i.e., the enzymes most responsible for cell destruction during the process of programmed cell death [46]. It has been proven that GAGs reduce cell damage by inhibiting NF-κB activation and apoptosis [47]. Therefore, it can be concluded that the proteolytic–prooxidative degradation of the ECM components in children with biologically untreated JIA also impairs—co-determined by GAGs—antioxidant and anti-inflammatory mechanisms. The latter properties of GAGs are associated with the aforementioned decrease by the analyzed heteropolysaccharides of NF-κB translocation to the cell nucleus and reduction in the MAP (mitogen activated protein kinases) signaling pathway [47]. Thus, activation of the latter group of serine/threonine protein kinases (ERK, extracellular signal-regulated kinase, and/or p38) in articular chondrocytes, determines—dependent on the influence of the proinflammatory cytokine IL-1β—the expression of MMP-1 and MMP-3 [48,49]. Thus, specific inhibitors of ERK (PD98059) and/or p38 (SB203580) and, probably, GAGs, reduce the stimulatory effect of IL-1β on the expression of MMPs [50]. The anti-inflammatory effect of GAGs is also associated with inhibition of the expression and activity of proinflammatory enzymes and the reduction in the synthesis of proinflammatory cytokines, including TNF-α or IL-1β. As a result, not only are the degradation processes of ECM components inhibited but also their synthesis is restored [51]. In consideration of the above analysis of the literature, as well as the significant inverse relationship found in the present study between plasma GAGs levels and the activity of the inflammatory process expressed by serum CRP levels in patients with JIA before the initiation of biological therapy, it seems that proinflammatory factors are the main modulators of GAGs metabolism in the course of JIA.

Apart from intensified catabolic processes, the dysfunction of the ECM components in the course of JIA may result from the impaired biosynthesis of these components. Among the factors significantly affecting the synthesis of the ECM components is TGF-β1 [52]. As reported in the previous study, the concentration of TGF-β1 in the blood of children with an aggressive form of JIA is significantly higher compared to the concentration characterizing healthy individuals [14]. Although we have not demonstrated a direct link between the glycan metabolism of matrix components and TGF-β1, the role of this factor in the regulation of ECM homeostasis cannot be ignored. The results of TGF-β1 concentrations in the blood of patients with JIA treated with etanercept indicate that ETA, besides leading to remission of the disease, simultaneously decreases the concentration of the assessed factor and, thus, returns TGF-β1-dependent homeostasis of chondrocytes and cartilaginous ECM. The effects of TGF-β1 on the alteration of the ECM of the cartilage, depending on its concentration, were described in our earlier work [14].

The mechanisms of ECM remodeling of cartilage tissue in the course of JIA indicate a significant stimulatory role of proinflammatory factors in this process. Proinflammatory factors lead to the inhibition of proteoglycan core protein synthesis in chondrocytes [53]. Deficiency of the core protein impairs the biosynthesis of glycosaminoglycan chains, which is reflected by low concentrations of GAGs in the blood of patients. However, the blood levels of GAGs reported in the present study, in patients after two years of ETA treatment, indicate that the pathways of transformation of the ECM components in children with an aggressive form of arthropathy requiring anticytokine therapy are complex. Indeed, ETA therapy contributing to clinical improvement in children led to significant changes in blood concentrations of GAGs, MMP-1, MMP-3, TOS, TAS, and TGF-β1. However, only in the case of TOS was its normalization observed. In other cases, the concentrations of the assessed markers were still significantly different in relation to the values reported in the blood of healthy children. It should be mentioned that one of the effects of biological therapy with ETA in children with JIA is the reduction in the pro-destructive effects of TNF-α on bone and joint components. TNF-α induces the synthesis of compounds that stimulate the regenerative processes of cartilage tissue, i.e., platelet-derived growth factor BB, but on the other hand, it is able to reduce the anabolic effect of other molecules, i.e., insulin-like growth factor 1 and TGF-β, which results in the domination of catabolic processes within the cartilage ECM [54,55]. The dynamics of changes in blood concentrations of GAGs in patients treated with ETA allowed us to conclude, initially to the sixth month of therapy, that there was an increase in the blood concentrations of GAGs, which may indicate increased biosynthesis of these compounds in tissues. However, it seems that the rebuilding of ECM should be accompanied by increased accumulation of GAGs in ECM, reflected by a decrease in the concentration of these compounds in blood. The above mechanism was observed in patients during the following months of treatment, indicating ECM regeneration. The dynamics of changes in MMP-1 and MMP-3 concentrations, similar to those of GAGs, confirm the interdependence of mechanisms regulating ECM metabolism. Significant associations between GAGs and MMPs confirm the role of proteolytic degradation of PGs in cartilage matrix remodeling in the course of JIA. The lack of normalization of GAGs blood concentrations seems to indicate that autoimmune disorders underlying the development of JIA may contribute to skeletal dysfunction in children.

## 5. Conclusions

The profile of changes in plasma concentrations of GAGs and modulators of their metabolism in blood of children with JIA before and during biological treatment seems to indicate that ETA therapy, although alleviating symptoms of JIA by reducing both pain and inflammatory processes, does not lead to the complete regeneration of the ECM elements damaged by the proteolytic and oxidative processes, which is reflected by the blood concentrations of the MMPs and oxidative stress markers. Taking into account the destructive potential of MMPs or ROS and their high expression in the course of JIA, reducing the overproduction of these compounds in children with arthropathy should bring clinical benefits, which are associated with the normalization of the metabolism of the extracellular matrix components, and, thus, delays the onset and maintenance of disease symptoms. The results of the present study indicate the necessity of the constant monitoring of the dynamics of the destructive processes of the articular cartilage in children with JIA, for example, by measuring the concentration of GAGs. We suggest that GAGs may be a useful biomarker to assess the clinical status of the extracellular matrix of joints.

## Figures and Tables

**Figure 1 biomedicines-10-01845-f001:**
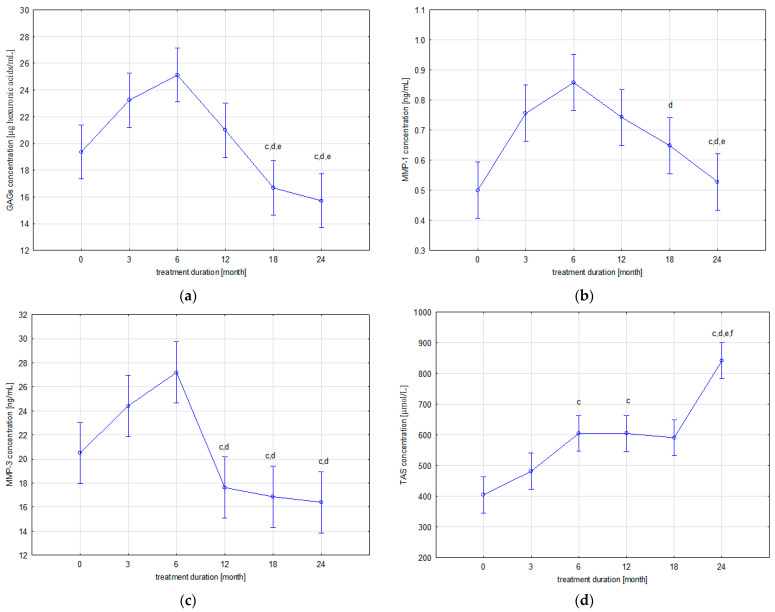
Dynamics of changes in plasma levels of GAGs (**a**), MMP-1 (**b**), MMP-3 (**c**), and TAS (**d**) in patients with JIA before and during etanercept therapy; ^c^ *p* < 0.05 compared to T3 group; ^d^ *p* < 0.05 compared to T6 group; ^e^ *p* < 0.05 compared to T12 group; ^f^ *p* < 0.05 compared to T18 group.

**Table 1 biomedicines-10-01845-t001:** The clinical data of control subjects and JIA patients.

Parameter	Control Subjects (*n* = 30)	JIA Patients (*n* = 38)					
		Before ETA Treatment T0	Time after Starting ETA Therapy				
			3 Months T3	6 Months T6	12 Months T12	18 Months T18	24 Months T24
**Age (years)**	8.01 ± 2.59	6.82 ± 2.04	7.04 ± 2.08	7.31 ± 2.07	7.82 ± 2.03	8.37 ± 2.71	8.84 ± 2.07
**Sex (F/M)**	21/9	25/10	25/10	25/10	25/10	25/10	25/10
**JADAS-27**	-	41.50 (36.50–49.50)	17.50 (15.50–21.50)	9.50 (8.00–13.50)	2.50 (1.00–4.00)	1.00 (1.00–1.50)	0.50 (0.00–1.00) ^b^
**Treatment Drugs**	-	MTX, EC, SSA	ETA, MTX, EC, SSA	ETA, MTX	ETA, MTX	ETA, MTX	ETA, MTX
**WBC (10^3^/μL)**	5.23 ± 2.15	9.88 ± 3.70 ^a^	7.07 ± 2.63	6.96 ± 2.85	6.75 ± 1.52	6.52 ± 1.62	6.28 ± 2.16 ^b^
**RBC (10^6^/μL)**	4.85 ± 0.33	3.87 ± 0.58 ^a^	4.51 ± 0.72	4.50 ± 0.82	4.48 ± 0.32	4.60 ± 0.39	4.86 ± 0.64
**Hb (g/dL)**	13.84 ± 1.81	11.35 ± 2.52 ^a^	11.99 ± 1.95	12.61 ± 4.40	13.50 ± 1.81	13.01 ± 1.46	13.80 ± 1.25 ^b^
**PLT (10^3^/μL)**	293.20 ± 71.56	348.95 ± 55.04	362.41 ± 53.88	318.95 ± 77.10	327.74 ± 84.96	312.05 ± 78.96	336.15 ± 50.50
**GPT (U/L)**	19.65 ± 7.98	23.96 ± 11.02	22.25 ± 7.41	17.56 ± 11.00	21.08 ± 8.30	24.45 ± 7.61	26.00 ± 10.08 ^a^
**GOT (U/L)**	25.68 ± 9.02	26.99 ± 10.98	26.70 ± 7.82	22.28 ± 7.41	22.19 ± 10.47	23.22 ± 10.87	25.92 ± 12.20
**Cr (mg/dL)**	0.69 ± 0.42	0.68 ± 0.55	0.70 ± 0.72	0.65 ± 0.23	0.70 ± 0.25	0.83 ± 0.25	0.96 ± 0.50 b
**ESR (mm/h)**	6.99 ± 2.21	42.85 ± 13.27 ^a^	29.41 ± 13.05	12.04 ± 7.84	8.95 ± 2.49	9.65 ± 6.60	8.87 ± 1.23 ^b^
**CRP (mg/L)**	0.67 (0.36–1.00)	23.83 (18.5–33.79)	13.98 (11.69–16.12)	0.79 (0.38–5.16)	0.74 (0.32–2.60)	0.45 (0.20–1.20)	0.43 (0.23–1.61) ^b^
**TOS (mmol/L)**	438.82 ± 140.96	1266.65 ± 526.77 ^a^	717.11 ± 356.63 ^b^	724.68 ± 300.56 ^b^	595.05 ± 301.87 ^b^	519.11 ± 277.00 ^b^	377.69 ± 160.66 ^b,c,d^
**TGF-β1 (ng/mL)**	6.94 ± 0.90	11.01 ± 2.28 ^a^	5.68 ± 2.10 ^b^	3.99 ± 1.41 ^b,c^	4.12 ± 1.26 ^b,c^	4.42 ± 1.54 ^b,c^	5.03 ± 1.37 ^a,b^

Results are expressed as mean ± SD or medians (quartile 1–quartile 3); ^a^ *p* < 0.05 compared to control group; ^b^ *p* < 0.05 compared to untreated JIA patients; ^c^ *p* < 0.05 compared to T3 group; ^d^ *p* < 0.05 compared to T6 group; ETA, etanercept; F/M, female/male; JADAS-27, Juvenile Arthritis Disease Activity Score-27; MTX, Methotrexate; EC, Encorton; SSA, Sulfasalazin; WBC, white blood cell; RBC, red blood cell; Hb, hemoglobin; PLT, platelet; GPT, glutamic pyruvic transferase; GOT, glutamic oxaloacetic transaminase; Cr, creatinine; ESR, erythrocyte sedimentation rate; CRP, C-reactive protein; TOS, total oxidative status; TGF-β1, transforming growth factor β1.

**Table 2 biomedicines-10-01845-t002:** The distribution patterns of plasma GAGs, MMP-1, MMP-3, and TAS in the healthy individuals (control subjects) and JIA patients.

Parameter	Control Subjects (*n* = 30)	JIA Patients (*n* = 38)					
		Before ETA Treatment T0	Time after Starting ETA Therapy				
			3 Months T3	6 Months T6	12 Months T12	18 Months T18	24 Months T24
GAGs (µg hexuronic acids/mL)	23.93 ± 2.40	19.37 ± 5.14 ^a^	23.24 ± 6.47	25.12 ± 8.84 ^b^	20.99 ± 5.60	16.67 ± 5.47	15.71 ± 5.83 ^a^
MMP-1 (ng/mL)	0.35 ± 0.10	0.50 ± 0.24 ^a^	0.75 ± 0.32 ^b^	0.86 ± 0.35 ^b^	0.74 ± 0.31 ^b^	0.65 ± 0.29	0.53 ± 0.23 ^a^
MMP-3 (ng/mL)	12.95 ± 3.39	20.50 ± 7.62 ^a^	24.42 ± 8.45	27.20 ± 11.79 ^b^	17.62 ± 7.40	16.86 ± 5.28	16.40 ± 5.70 ^a^
TAS (µmol/L)	726.25 ± 179.04	405.00 ± 121.26 ^a^	481.66 ± 149.67	605.19 ± 175.84 ^b^	604.67 ± 215.65 ^b^	591.19 ± 258.06 ^b^	842.04 ± 149.76 ^a,b^

Data are expressed as mean ± standard deviation; ETA, etanercept; GAGs, glycosaminoglycans; MMP, matrix metalloproteinase; TAS, total antioxidative status; ^a^ *p* < 0.05 compared to control group; ^b^ *p* < 0.05 compared to T0 group.

**Table 3 biomedicines-10-01845-t003:** Correlation analysis between plasma GAGs and MMP-1, MMP-3, TAS, TOS, TGF-β1, and CRP levels in JIA patients.

Parameter	JIA Patients (*n* = 38)					
	Before ETA Treatment T0	Time after Starting ETA Therapy				
		3 Months T3	6 Months T6	12 Months T12	18 Months T18	24 Months T24
MMP-1 r(p)	0.768 (*p* = 0.000)	0.678 (*p* = 0.000)	0.738 (*p* = 0.000)	0.779 (*p* = 0.000)	0.692 (*p* = 0.000)	0.693 (*p* = 0.000)
MMP-3 r(p)	0.386 (*p* = 0.017)	0.614 (*p* = 0.000)	0.664 (*p* = 0.000)	0.701 (*p* = 0.000)	0.720 (*p* = 0.000)	0.811 (*p* = 0.000)
TAS r(p)	−0.865 (*p* = 0.000)	0.718 (*p* = 0.000)	−0.917 (*p* = 0.000)	−0.214 (NS)	−0.163 (NS)	−0.248 (NS)
TOS r(p)	−0.118 (NS)	0.753 (*p* = 0.000)	−0.080 (NS)	−0.791 (*p* = 0.000)	−0.175 (NS)	0.541 (*p* = 0.000)
TGF-β1 r(p)	0.222 (NS)	−0.031 (NS)	−0.065 (NS)	−0.024 (NS)	−0.180 (NS)	−0.233 (NS)
CRP r(p)	−0.769 (*p* = 0.000)	−0.257 (NS)	0.114 (NS)	0.439 (*p* = 0.006)	0.012 (NS)	−0.073 (NS)

Results are expressed as the Pearson’s correlation coefficients (r); ETA, etanercept; MMP, matrix metalloproteinase; TAS, total antioxidative status; TOS, total oxidative status; TGF-β1, transforming growth factor β1; CRP, C-reactive protein; NS, not statistically significant.

## Data Availability

The datasets analyzed or generated during the study are available from the lead author: magdalena.bacia@gmail.com.

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
