# Peer review of "Plasma Glycosaminoglycans in Children with Juvenile Idiopathic Arthritis Being Treated with Etanercept as Potential Biomarkers of Joint Dysfunction"

_biomedicines, 2022, doi:10.3390/biomedicines10081845_

Round 1
Reviewer 1 Report
The present study aims to identify a useful biomarker to monitor the clinical status of the ECM of joints of patients with juvenile idiopathic arthritis. The authors quantified GAGs concentration in blood, together with that of MMP-1 and MMP-3, and of total antioxidants, for 24 months, in a cohort of children with JIA treated with etanercept. They have found that GAGs might be used as a marker of joint dysfunction and indicators of biological effectiveness of therapy.
The work is interesting, and the manuscript is well-written. Methods are clearly described, and results are well presented and discussed. However, there are several points needing correction and/or revision, as follows.
1. Line 45: “of a joint”.
2. Lines 49-51: it is better to revise to “The ECM is a multicomponent, dynamic network structure with gel-like properties, especially in cartilage, whose composition and arrangement determine its mechanical functions and immunological characteristics”.
3. Line 72: delete “also”.
4. Line 128: “SSD” (two times) should be “SSA”.
5. Table 1: “SSD” (two times) should be “SSA”; correct the numbers in WBC, RBC and PLT, as well as the numbers in CRP concentration row.
6. Lines 170 & 173: correct the number of temperatures.
7. Lines 219 & line 228: GAGs, MMP-1 & MMP-3 concentrations increased the first 6 months. This should be also discussed.
8. Line 371: Other proteases, like ADAMTS, or other MMPs, do participate in the depletion of aggrecan. Since the same group has presented earlier this year results regarding the concentration of ADAMTS-4 and -5 in the blood of children with JIA treated with ETA (ref 14), it will be interesting to discuss (or better to examine) any relation between GAGs and ADAMTS enzymes concentration.
Reviewer 2 Report
In this study, the authors continuously measured the levels of GAGs as well as the level of MMPs related to the degradation of PGs/GAGs and the level of oxidative stress-related markers in the plasma of children with Juvenile Idiopathic Arthritis. By analyzing its correlation, it is proposed that GAGs in plasma can be used as a marker for monitoring changes in the extracellular matrix of joints. However, there are a number of points need clarifying and certain statements require further justification.
1. Page 3, line 107, the abbreviation of sulfasalazine is SSA,but in subsequent use, for example, in the line 128 it was SSD, and in the Table1 the part of treatment drugs is shown SSD, however in line 155 and line 363 were SSA. Please check to make it clear whether the SSA and SSD mentioned in the article were the same drug, and if they are the same, the abbreviation of sulfasalazine should be consistent.
2. In the part of patients and sample, line 106-109, the authors mentioned that MTX, SSA, EC treatment failed to improve clinical status or not tolerated were qualified for the study. However, in line127-129,“ETA was used together with the MTX, SSD and EC. After 3 months of effective therapy, both SSD and EC were withdrawn.” What is the reason or basis for such administration? Please explain.
3. For the detection of oxidative stress, the authors tested the total antioxidant stress markers and oxidative stress markers. There are many kinds of oxidative stress-related enzymes including glutathione peroxidase(GSH-Px), Superoxide dismutase(SOD); Catalase(CAT)et al. The authors can refine the detection of these markers, and perhaps have other finding and further understand the relationship between the disease and oxidative stress.
4. The content of Table 1 is mainly introduced in the materials and methods. It is recommended to add a description of Table 1 in the results, because CPR, TOS and TGFβ- are found by comparing the healthy subjects and JIA patients before and after treatment and carry out follow-up experiments to explain scientific problems.
5. Whether other markers are present during joint remodeling. Figure 1, in the treatment process of T0-T24, in addition to MMPs and antioxidant stress markers, it is recommended to increase the detection of direct markers representing joint remodeling after treatment to indicate joint changes, such as cartilage-related markers, to detect GAGs and markers correlations of other substances to further increase the evidence for GAGs as a bio-marker representing joint state.
